# A Comprehensive Research on the Prevalence and Evolution Trend of Orthopedic Surgeries in Romania

**DOI:** 10.3390/healthcare11131866

**Published:** 2023-06-27

**Authors:** Flaviu Moldovan, Liviu Moldovan, Tiberiu Bataga

**Affiliations:** 1Orthopedics—Traumatology Department, Faculty of Medicine, George Emil Palade University of Medicine, Pharmacy, Science, and Technology of Targu Mures, 540142 Targu Mures, Romania; tiberiu.bataga@umfst.ro; 2Faculty of Engineering and Information Technology, George Emil Palade University of Medicine, Pharmacy, Science, and Technology of Targu Mures, 540142 Targu Mures, Romania; liviu.moldovan@umfst.ro

**Keywords:** orthopedic surgeries, hip replacement surgery, knee replacement surgery, revision burden, National Endoprosthetic Registry

## Abstract

Accurate knowledge of the prevalence and trends of orthopedic surgeries can facilitate the design of medical plans for effective treatments. The National Endoprosthetic Registry (NER) in Romania provides statistics on endoprosthetic activity (hip, knee), cases of fractures and bone tumors as a result of the legal obligations to report interventions performed by all orthopedic traumatology hospitals/wards in the country. The aim of this study is to describe the annual volumes of orthopedic surgeries between 2001 and 2022 in Romania and analyze the current and future evolution trends of the studied surgeries, gender differences and regional differences based on a complete survey carried out at a national level. For the period 2001–2022, we extracted from the NER the annual volumes of orthopedic interventions performed. With these data, we studied the prevalence and estimated, with the support of an original calculation methodology, the variation trends of orthopedic surgeries in two situations: over the entire 21-year period, respectively, and over the period 2001–2020, which does not include the pandemic period. For hip replacement surgery and knee replacement surgery, we showed the prevalence by subcategory of interventions, gender distribution, regional prevalence and regional density calculated by the annual averages of the total number of cases reported per 100,000 people in the 40 counties of the country and the capital, Bucharest. We also determined the variations in hip and knee arthroplasty revision burdens, calculated as a percentage between the number of revisions and the number of primary interventions in the same period. We determined the regional densities of revision burdens. The total number of orthopedic surgeries in the period 2001–2022 was 1,557,247, of which 189,881 were hip replacement surgeries; 51,035 were knee replacement surgeries; 11,085 were revision hip arthroplasty; 1497 were revision knee arthroplasty; 541,440 were operated fractures; and 16,418 were operated bone tumors. The growth rates of surgical interventions are hip replacement surgery, +8.19%; knee replacement surgery, +19.55%; revision hip arthroplasty, +9.43%; and revision knee arthroplasty, +28.57%. With these data, we have estimated a doubling of the volume of primary and revision interventions of the hip until 2034 and the knee until 2027, respectively. Operated bone tumors register an annual decrease of −4.52% thanks to modern treatments. There are clear gender differences; for primary hip interventions, the proportion of women is 58.82%, and for knee interventions, the proportion of women is 76.42%. This is the first research that, with the support of exhaustive data from the NER, analyzes for the period 2001–2022 the annual number of orthopedic surgeries in Romania. It allows knowledge of the large, anticipated increases in orthopedic surgery and provides a quantitative basis for future policy decisions related to the need for medical personnel and material resources.

## 1. Introduction

Arthroplasty is an effective treatment for patients experiencing permanent restriction of joint function due to joint destruction that cannot be treated by other methods. Hip and knee arthroplasties can restore joint function, weight-bearing capacity and quality of life [1]. Accurate knowledge of the prevalence and trends of arthroplasties facilitates the design of medical plans for effective treatments [2,3]. For this, large studies can be carried out with the support of national arthroplasty registries, which are now available in several countries and which provide accurate information on the number of annual arthroplasties [4,5]. 

In Romania, through the National Public Health Program called “Prevention in Orthopedics and Traumatology” of the Ministry of Health, the National Endoprosthesis Register (NER) was launched in 2001 [6]. The main purpose of the register is to act as a surveillance tool for the volumes of orthopedic surgery performed nationally. It also constitutes its own database for each orthopedic service, where it can see all its reported data that it can compare with the totalized volumes per country. The registry operates on a non-competitive basis and patient information is confidential. All orthopedic traumatology hospitals/wards are obliged to report to the NER the activity of endoprosthesis (hip, knee) and cases of fractures and bone tumors in accordance with the provisions of Order 1591/1110/2010 [7]. At the moment, 125 hospitals are registered in the register, and a complete investigation is possible regarding three categories of interventions: national situations about hip replacement, national situations about knee replacement and national situations about tumors and fractures in orthopedic services. By reporting the results, the information in the registry allows comparisons with other registries, economic analyses [8,9,10], estimation of changes in the number of procedures and the use of resources over time [11,12].

A number of studies indicate constant demands for total hip and knee arthroplasties [11,13], and these demands will increase over the next 10 years in most age groups [14]. Although recovery after arthroplasties is facilitated by the application of protocols such as Enhanced Recovery After Surgery for primary hip and knee arthroplasty [15,16,17], as well as the provision of basic elements in improvement pathways [18], there is an increase in the incidence of the total hip and knee revision procedures [19]. At the same time, the number of operated fractures [20,21] and bone tumors [22,23] are also increasing. All this will lead to an increase in healthcare expenses in the future [24,25], even during periods of economic recession [10,26].

In the USA over the past thirteen years, the rate of total hip arthroplasties increased by approximately 50%, and the rate of total knee arthroplasties nearly tripled [27]. Additionally, during one decade in the USA, the revision rate of total hip arthroplasties increased by 3.7%, and the revision rate of total knee arthroplasties increased by 5.4% [27].

**Hypothesis** **1** **(H1).***There is a trend of annual increase in the prevalence of orthopedic surgeries in Romania, and the annual rate of growth can be predicted*. 

Studies show that in the case of hip arthroplasties, the proportion of female cases was almost 83%, and in the case of knee arthroplasties, the proportion of female cases was almost 80% [28]. Knowing the annual rates of increase in the volumes of surgical interventions for a certain population, the weights of interventions by gender are useful in estimating the volume of interventions in the following years. With this information, the need for medical personnel and hospital infrastructure in the coming period can be estimated.

**Hypothesis** **2** **(H2).***There are gender differences; the number of arthroplasties performed in women is higher than the number of arthroplasties performed in men, and knowing them allows predicting the effort of interventions in the following period*.

Studies show that although there are regional differences in the number of hip and knee arthroplasties by administrative units, their number in relation to the population shows a small variation between administrative units [28].

**Hypothesis** **3** **(H3).***The prevalence of hip and knee arthroplasties correlates with the population served by the hospitals in the administrative unit*.

The main measure to evaluate the result of arthroplasty is the revision operation. The most common reasons for revision are deep infection, aseptic loosening, instability, periprosthetic fracture, prosthetic insufficiency, excessive wear, stiffness or unexplained pain [29,30,31]. 

The concept of revision burden, defined as the ratio between the number of revision interventions and the total number of primary arthroplasties in a given period, is a measure of the steady state and success of arthroplasty. This concept is a simple, universally comparable parameter introduced by the director of the Swedish Hip Arthroplasty Registry (SHAR), Dr. Henrik Malchau [32], to facilitate comparison with other national registries. This method of calculation is not universally accepted [33,34,35], but it is frequently used for analyses and comparisons by other researchers as well [11,27].

Numerous studies [8,27,30] have shown that revision surgeries require more resources and have poorer durability and outcomes compared to the relative cost and outcomes of primary hip and knee replacement. If by comparing the results of the orthopedic departments, the revision tasks of the hip or the knee are significantly different, explanations for the observed differences can be requested. In this way, best practices can be highlighted that can promote favorable changes. This serves to strengthen the efficiency of medical care in high-performing systems and enables the improvement of systems with higher revision loads. 

**Hypothesis** **4** **(H4).***The revision burdens for hospitals located in the counties in Romania are close, and identifying the areas where the hospitals with the smallest revision burdens are located allows the promotion of best practices*.

Our specific aims were:-To describe the annual volumes of orthopedic surgical interventions in the period 2001–2022 and to estimate the trends of evolution, taking into account the statistical data recorded in the NER;-To analyze the temporal trends of primary endoprosthesis surgeries, gender differences and regional differences;-To analyze the current trends in the hip and knee revision burdens and to identify the regional performances;-To estimate the volumes of surgical interventions that are expected to be performed in the following years by the National Health Service in order to establish the need for specialized human resources and material endowments.

## 2. Materials and Methods

The methodology of this study consisted of:Study design and selection of study participants;Selection of the variables explored in the study and calculus methodology;Data collection and statistical analysis.

### 2.1. Study Design and Participants

The methodology of this study consisted of a retrospective analysis of the patients’ beneficiaries of primary and revision hip and knee surgery from 125 orthopedic traumatology hospitals/departments in Romania. All surgical interventions that were performed between 1 January 2001 and 31 December 2022 and that were reported by hospitals to the NER were extracted. All patients regardless of age and gender were included in the study.

For this, we used all data reported in NER for the following categories of interventions: Hip replacement surgery, detailed by subsequent surgeries: total hip arthroplasty—code O12104 (in turn, detailed into prostheses: cemented total hip arthroplasty, uncemented total hip arthroplasty, hybrid and reverse hybrid total hip arthroplasty), bipolar hemiarthroplasty, unipolar hemiarthroplasty—Moore type—code O12103. We also extracted the gender of the patients who underwent these interventions;Knee replacement surgery, detailed by subsequent surgeries: bicondylar knee arthroplasty—code O14902, unicondylar knee arthroplasty—code O14901. We also extracted the gender of the patients who underwent these interventions;Revision hip arthroplasty—code O12401;Revision knee arthroplasty—code O15501;Operated fractures, including some of the following: closed reduction of femur fracture with internal fixation (code O11808), open reduction of the ankle fracture with internal fixation of the diastasis, fibula or malleolus (code O16801), closed reduction of tibial diaphysis fracture with internal fixation (code O14002), open reduction of the radius and ulnar diaphysis fracture with internal fixation (code O06504), closed reduction of the humeral shaft fracture with internal fixation (code O04904), etc.;Operated bone tumors, including some of the following: benign bone tumor resection with anatomically specific allograft (code: O19009 ), marginal excision of the malignant bone tumor with cementation of the defect (COD: O19605), en bloc resection of the malignant long bone tumor of the lower limb with arthrodesis of the adjacent joint (COD: O19703), en bloc resection of the malignant long bone tumor of the upper limb with replacement of the adjacent joint (COD: O19702), etc.;Hospitalized patients;Operated patients.

A total of 1,557,247 subjects were included in the study, which were registered in NER (Table 1).

Patient consent was obtained for data collection and inclusion in NER. According to the specifications of the Ministry of Health, separate informed consent and ethical approval were not required for the present study.

### 2.2. Evaluation Variables and Methodologies

Based on the formulated hypotheses, we have defined the variables and have developed the evaluation methodologies appropriate to the following explored issues: the volumes of surgical procedures, gender differences, regional relevance and revision burden.

#### 2.2.1. The Volumes of Surgical Procedures and Growth Rates

In order to estimate the tendency of volumes of surgery procedures, we selected as variables the annual volume numbers of the following interventions: hip replacement surgery (nHrs); total hip arthroplasty (nTHA); cemented total hip arthroplasty (nCTHA); uncemented total hip arthroplasty (nUTHA); hybrid and reverse hybrid total hip arthroplasty (nHTHA); bipolar hemiarthroplasty (nBH); unipolar hemiarthroplasty—Moore type (nUH); hip replacement surgery—male (nHrsM); hip replacement surgery—female (nHrsF); knee replacement surgery (nKrs); bicondylar knee arthroplasty (nBKA); unicondylar knee arthroplasty (nUKA); knee replacement surgery—male (nKrsM); knee replacement surgery—female (nKrsF); revision hip arthroplasty (nRHA); revision knee arthroplasty (nRKA); operated fractures—upper limb, lower limb, pelvis (nOf); operated bone tumors—upper limb, lower limb, pelvis (nObt); hospitalized patients (nHp); operated patients (nOp).

We estimated the trends by calculating the arithmetic mean of the percentage values of the differences in the volumes of interventions from two successive years y + 1 and y, related to the value of the volume of interventions from the base year y for all 20 categories of interventions studied.

In the case of hip replacement surgery, the variation trend is expressed in the form:(1)$$VT=\sum_{y=1}^{Y}\frac{nHrs\left(y+1\right)-nHrs(y)}{Y\times nHrs(y)}\cdot 100[\%]$$
where nHrs(y) is the number of hip replacement surgeries related to year “y” from the interval 2001−2022. Similar formulas were used for the other categories of interventions.

#### 2.2.2. Gender Differences

In order to evaluate the gender differences in performing arthroplasties, we selected as variables the ratio between the numbers of interventions performed in women/men over the total number of interventions ×100 for the categories of interventions presented in Section 2.2.1.

#### 2.2.3. Regional Prevalence

In order to evaluate the prevalence of hip and knee arthroplasties in correlation with the number of the population served by the hospital, we selected as variables the annual ratio between the volume number of the interventions presented in Section 2.2.1 and the number of the population per county (NP_county1...41_).

#### 2.2.4. Revision Burden

For the study of revision burdens, we selected as a variable the percentage of arthroplasty revisions, which is determined as the ratio of the number of revision arthroplasties/the number of primary arthroplasties in the same time period ×100. In this way, we calculated the hip revision burden
(2)$$HRB=\frac{nRHA}{nTHA}\cdot 100[\%]$$
and knee revision burden, respectively
(3)$$KRB=\frac{nRKA}{nTKA}\cdot 100[\%]$$

We explored the revision burdens for the counties in Romania by accumulating the interventions carried out by all the hospitals located in the respective administrative units.

By describing the proportions of hip and knee arthroplasties that required revision surgery, surgeons have insight into the reliability of the interventions performed.

### 2.3. Data Collection and Statistical Analysis

In February 2023, data were collected from the NER that we obtained from the “Useful information” section on the National Endoprosthesis Registry website. From the presented linear and column graphs, which indicate the number of cases per year, we extracted in Excel files the volumes related to the categories of interventions studied. For this, we examined the annual trend in the number of surgeries corresponding to the period 2001–2022 (Appendix A).

Gender-specific values were collected for hip replacement surgery (nHrs) and knee replacement surgery (nKrs) interventions. For the other subcategories of interventions, no gender-specific values were collected, as they are not presented in the RNE.

Next, we extracted from the section “Orthopedic services of the RNE” the specific numbers of interventions performed by the 125 hospitals that report to the NER. We accumulated the specific numbers of hospital interventions in the same county, corresponding to the 40 counties and the municipality of Bucharest, which we reported per 100,000 inhabitants of each county and Bucharest, respectively.

We obtained the estimated population by county and the distribution by gender from the National Institute of Statistics [36] from the table: “Population by gender, by residence—counties and averages”. 

The data were filtered, analyzed primarily and transferred to Microsoft Excel, GNU PSPP and Matlab for further processing. Statistical analysis was performed with Statistics and Machine Learning Toolbox Version 12.3 from Matlab R2022a (The MathWorks, Inc., Natick, MA, USA).

## 3. Results

In our study, we addressed the data of the 125 orthopedic traumatology hospitals/departments in Romania, which are obliged to report their joint replacement surgery activity (hip, knee) to the National Endoprosthesis Register [6] by Order of the Minister of Health. No hospital was excluded from the list. It represents a proportion of 100% of the orthopedic traumatology hospitals/departments existing at national level.

### 3.1. All Types of Orthopedic Surgeries

With the support of the data from Table 1, we represented in Figure 1 the percentages of orthopedic surgeries performed in the period 2001–2022. The analysis shows that the largest share of the total orthopedic surgeries from the studied period 2021–2022 is operated fractures (66.73%), followed by hip replacement surgery (23.40%), knee replacement surgery (6.29%), operated bone tumors (2.02%), revision hip arthroplasty (1.38%) and revision knee arthroplasty (0.18%).

In the continuation of the study, we examined the annual trend in the volume of cases between 1 January 2001 and 31 December 2022 for hip replacement surgery, knee replacement surgery, revision hip arthroplasty, revision knee arthroplasty, operated fractures and operated bone tumors (Figure 2).

We calculated the variation trends of the surgical interventions studied with formula (1) for a number of Y = 21 years corresponding to the interval (2001−2022), which also includes the pandemic period (Table 2). We performed the same calculations for the period of Y = 18 years (2001−2019) without the pandemic period. Numerical values for which no finite numbers were obtained were removed from the calculation because they have no physical meaning.

As shown in Figure 2 and Table 2, the highest annual growth rates in descending order are for revision knee arthroplasty (RKA), 28.57%; knee replacement surgery—male (KrsM), 20.87%; bicondylar knee arthroplasty (BKA), 20.41%; and knee replacement surgery—female (KrsF), 19.57%. It is worth noting that there are interventions with decreasing rates: operated bone tumors (Obt)—4.52%, operated patients (Op)—1.89% and hospitalized patients (Hp)—0.56%.

This partially confirms the H1 hypothesis that there is a trend of an annual increase in the prevalence of orthopedic surgeries in Romania. By calculating the annual rate of growth, we demonstrated that some interventions, such as revision knee arthroplasty, are increasing and others, such as operated bone tumors, are decreasing. We have forecasted the annual growth rate for the studied interventions, which is applicable for the immediate future, but as data are collected in the NER, this can be updated for the highest accuracy predictions.

### 3.2. Hip Replacement Surgery

Next, we studied the annual trend between 1 January 2001 and 31 December 2022 in the volume of cases for the interventions that make up hip replacement surgery. In Figure 3a, we represented the interventions that make up total hip arthroplasty: cemented total hip arthroplasty, uncemented total hip arthroplasty and hybrid and reverse hybrid total hip arthroplasty, as well as intervention bipolar hemiarthroplasty and unipolar hemiarthroplasty—Moore type. In Figure 3b, we represented the cumulative gender distribution of the interventions that make up hip replacement surgery.

As can be seen from Figure 3a and Table 2, in these intervention categories, the highest annual growth rate is uncemented total hip arthroplasty (15.22%), while unipolar hemiarthroplasty (0.07%), bipolar hemiarthroplasty (0.12%) and hybrid and reverse hybrid total hip arthroplasty (0.22%) are approximately constant over time. In the period 2001−2022, 189,881 cases were registered, of which 78,190 patients were men, i.e., 41.18%, and 111,691 patients were women, i.e., 58.82%.

This confirms hypothesis H2 in the case of hip replacement surgery according to which the number of arthroplasties performed in women is 17.64% higher than the number of arthroplasties performed in men. The knowledge of these proportions, in conjunction with the variation trends of the surgical interventions calculated in Section 3.1, allows forecasting the effort of hip arthroplasty interventions in the following period by referring to the number of the population in the administrative units of the country.

### 3.3. Knee Replacement Surgery

The annual trend between 1 January 2001 and 31 December 2022 in the volume of cases for the interventions that make up the knee replacement surgery interventions bicondylar knee arthroplasty and unicondylar knee arthroplasty are represented in Figure 4a. In Figure 4b, we represented the cumulative gender distribution of the interventions that make up knee replacement surgery.

As can be seen from Figure 4a and Table 2, in the case of knee replacement surgery, the annual growth rate is 19.55%. The highest annual growth rate is bicondylar knee arthroplasty (20.41%). In the period 2002−2022, 51,035 cases were registered, of which 12,036 patients were men, i.e., 23.58%, and 38,999 patients were women, i.e., 76.42%.

This confirms hypothesis H2 in the case of knee replacement surgery, according to which the number of arthroplasties performed in women is 52.84% higher than the number of arthroplasties performed in men. The knowledge of these proportions, in conjunction with the variation trends of surgical interventions calculated in Section 3.1, allows forecasting the effort of knee arthroplasty interventions in the following period by referring to the number of the population in the administrative units of the country.

### 3.4. Regional Prevalence

The total number of cases from the interval 2002–2022 for the 40 counties and the capital Bucharest are presented in Figure 5a. The annual average of the number of cases is represented in Figure 5b.

The data analysis shows that for the studied interval, in the case of hip replacement surgery (Figure 5a), the largest volumes of interventions are in Bucharest (53,498 cases), Mures (19,257 cases), Timis (9096 cases), Cluj (8854 cases) and Brasov (8557 cases). The annual average of the number of cases in descending order (Figure 5b) is recorded in Bucharest (2431.73 cases), Mures (875.32 cases), Timis (413.45 cases), Cluj (402.45 cases) and Brasov (388.95 cases).

In the case of knee replacement surgery (Figure 5a), the highest cumulative numbers of cases are in Bucharest (22,867 cases), Mures (3761 cases), Timis (3438 cases), Cluj (2964 cases) and Bihor (2261 cases). The annual average of the number of cases in descending order (Figure 5b) is registered in Bucharest (1088.90 cases), Mures (188.05 cases), Timis (163.71 cases), Cluj (148.20 cases) and Bihor (113.05 cases).

In Figure 6, maps of the densities of hip and knee arthroplasties are represented by the annual averages of the total number of cases reported per 100,000 people in the 40 counties and the capital Bucharest for the period 2001–2022. The highest densities of hip arthroplasties (Figure 6a) are registered in Mures (150.70), Bucharest (126.24), Brasov (104.23), Timis (60.99) and Cluj (57.24). According to the representation in Figure 6b, the highest densities of knee arthroplasties are recorded in Bucharest (53.96), Mures (29.43), Brasov (925.77), Timis (23.05) and Cluj (17.19).

From the analysis of these data, it can be concluded that hypothesis H3 is not confirmed, because there are large variations in the density of arthroplasties in relation to the number of the population served by hospitals in different counties. The minimum values of the density of hip arthroplasties have the lowest values in the counties of Tulcea (0.04), Teleorman (2.02) and Buzau (3.14). Additionally, knee arthroplasties have extremely low densities in counties such as Buzau, Calarasi and Vrancea.

Given the fact that the counties where the densities of arthroplasties are higher are university medical centers, it can be concluded that university hospitals serve a larger share of the population. Romania has 12 university medical centers, and among them, the most active in the field of primary endoprosthesis interventions are Bucharest, Targu Mures, Cluj–Napoca, Timisoara and Brasov.

### 3.5. Revision Burden

The results regarding the hip and knee arthroplasty revision burdens at the national level are presented in Table 3, and their variations are represented in Figure 7.

Revision burdens were 5.40–7.23% at the hip joint and 1.46–4.33% at the knee joint, respectively. The increasing trend in the number of revision interventions was manifested mainly in the period 2001–2013, after which they registered decreases, especially during the pandemic.

In Figure 8, maps of the revision burdens are represented in the 40 counties and the capital Bucharest for the period 2001–2022. The highest revision burden of the hip is registered in Mures (0.33%), Bucharest (0.21%), Brasov (0.19%), Cluj (0.08%) and Timis (0.06%). Knee revision records the highest revision burden in Bucharest (3.89%), Mures (2.03%), Timis (1.87%), Sibiu (1.65%) and Brasov (1.25%).

This finding does not confirm hypothesis H4 because there are consistent variations in revision burdens between hospitals located in different counties of the country. The highest revision burdens are in the university centers in Targu Mures, Bucharest, Brasov, Cluj–Napoca, Timisoara and Sibiu. We also appreciate that the other regional hospitals, which have smaller revision burdens, do not have university medical staff, and as a result, hypothesis H4 cannot be confirmed, according to which they have better surgical practices.

## 4. Discussion

For the 22-year period between 2001 and 2022, the growth rate of hip replacement surgery was +8.19%. In the same period, the country’s population decreased from 21.6 million inhabitants in 2002 to 19 million inhabitants in 2022, and the share of the population aged 65 and over increased by 4.3%, from 14% to 19.6% [36]. The rate of increase in the number of hip replacement surgeries was approximately double the increase in the elderly population, which may be due to the expansion of indications for these interventions according to the good clinical results [37]. This growth trend could lead to a doubling of the number of hip replacement surgeries in Romania by 2034. Our findings are consistent with the results of the study published by Pabinger and Geissler [38], who showed that hip arthroplasty is increasing exponentially in OECD (Organization for Economic Co-operation and Development) countries, but the rates vary from one country to another: USA, +12.87%; Australia, +7.77%; United Kingdom, +6.95%; Germany, +6.07%; Spain, +6.73%; etc.

The growth rate of the number of knee replacement surgeries for the studied period was +19.55%. This is mainly due to the recommendations for surgical intervention in the diagnosis of painful knee osteoarthritis [39] and to the good clinical outcomes after a knee replacement as a result of the application of rapid recovery protocols [40]. With this growing trend, the volume of knee replacement surgery will double in Romania in the next 5 years, i.e., until 2027. Our findings are consistent with the results of the study by Pabinger et al. [41], who showed that knee arthroplasty is growing exponentially in OECD countries due to the prevalence of obesity: USA, +20.26%; Australia, +18.46%; United Kingdom, +13.06%; Germany, +12.80%; Spain, +11.76%; etc. Higher rates of increase are seen in patients aged 64 years or younger compared to the older population. This trend could lead to a fourfold increase in demand for knee arthroplasty in OECD countries by 2030.

For knee arthroplasty interventions, we found 2.38 times higher growth rates compared to those for the hip. Our finding is consistent with the results of the study conducted by Falbrede et al. [42] in Germany, Switzerland and the USA, which also reported annual increases in the rates of primary arthroplasties, with knee arthroplasties being higher. This situation is undoubtedly due to the failure of some new technologies that recorded poor results, such as the ASR XL Acetabular System and the ASR Hip Resurfacing System [43], which were withdrawn by the manufacturer from the worldwide market, as well as the components fabricated from cobalt–chromium alloy [44] or due to corrosion at the modular neck–stem junction systems [35].

Operated bone tumors recorded a decrease of −4.52%, probably due to chemotherapy treatments, radiation therapy, as well as novel therapeutic approaches and hydrogel-based anticancer therapy [45]. Additionally, new therapeutic methodologies, nanotechnology-based anticancer therapy [46] and bifunctional biomaterials combining tumor photothermal therapy with enhanced bone regeneration [47] were applied to patients with osteosarcoma.

Clear gender differences were evident for primary arthroplasty interventions, as the proportion of women who had hip replacement surgery was 58.82%, and the proportion of women who had knee replacement surgery was 76.42%. This finding is consistent with other studies that indicate female proportions of up to 70–90% for all-joint arthroplasties [28,48,49,50].

The distribution of the number of primary arthroplasty interventions by county, related to the population, showed consistent variations. The densities of hip replacement surgery and knee replacement surgery, respectively, recorded the highest values in the capital Bucharest and the counties of Mures, Brasov, Timis and Cluj. These are among the 12 counties where there are university medical centers. We found that the distribution of arthroplasties performed in regional hospitals is not proportional to the share of the population, with hospitals located in university centers being preferred. However, larger centers do not report more cases, with special regional performances highlighted for Mures, which has the highest densities of hip arthroplasties and which, for certain categories of interventions, is ahead of other larger university centers, such as Bucharest, Cluj, Timis and Brasov. This finding is somewhat different from the result of the study conducted in Japan by Katano et al. [28], who indicated small variations in the densities of arthroplasties per population and prefectures. 

Revision burden is a measure that may be useful in tracking the effect of changes in surgical technique and implant design over time on the registry population. The hip replacement surgery revision percentages (7.23%) have higher values compared to the knee replacement surgery revision percentages (4.33%). However, the results of this analysis should be interpreted with caution because revision rates were calculated as the ratio of the number of revisions over the number of primary interventions performed in the same year. The accuracy of revision rates would be more relevant if patients were followed continuously. The relative decreases in revision burdens after 2013 seem to be due to improvements in implant design and materials used, such as highly cross-linked polyethylene, but also feedback from surgeons, hospitals and manufacturers [51,52].

The +9.43% growth rate of the number of hip arthroplasty revisions predicts a doubling of the volume of interventions by 2034. The growth forecast for revision knee arthroplasty of +28.57 will lead to a doubling of the volume of interventions in less than 4 years. We found 3.02 times higher growth rates for knee revisions compared to hip revisions. 

The distribution of the number of revision interventions by county, related to the number of the population, showed consistent variations, and the highest densities of hip and knee revisions are recorded in the university centers that perform the largest volumes of primary interventions. Larger centers do not report more cases, and special regional performances are in Mures, which has the highest revision burden of the hip.

However, comparing hip replacement surgery and knee replacement surgery shows a much higher number of hip replacement surgeries, 189,881, versus 51,035 knee replacement surgeries. The density per county related to the population has an average value of 30.72 for interventions on the hip and 6.60 for interventions on the knee, respectively. This suggests a more stable surgical indication for the hip joint. Additionally, the indication for surgical intervention in each county is more constant for the hip than for the knee. This result must be analyzed, taking into account the findings of the study conducted by Sabah et al. [53], who are of the opinion that due to underreporting and lack of data on implant revisions and failure rates, they are more vulnerable compared to primary arthroplasty procedures.

The first limitation of this study questions the accuracy of data from the National Endoprosthetic Registry and the percentage of interventions that are reported. According to national legislation, all hospitals that perform orthopedic surgeries must report them periodically to the NER. However, there may be delays in the reports sent by some hospitals, so the totals are periodically adjusted for records from the last 1–2 years. Another limitation is due to the lack of records of patients’ personal data, which would allow a continuous follow-up of them. NER does not provide data on diagnosis, surgical procedures and implant models or other information such as the type of implanted socket (hemispherical, threaded), the type of stem (surface replacement, short metaphyseal, standard), type of articulation (ceramic/ceramic, metal/polyethylene, etc.), the cause of revision, etc. The information regarding the age and domicile of the patients is also not known. The calculation mode of the revision tasks does not use the direct links between the initial operation and the revision one, and for this reason, a survival analysis of the implants cannot be performed. The National Endoprosthetic Registry offers the possibility to observe trends only in the most elementary domains without allowing the exploration of more precise domains.

Future research directions should include in the analysis the updated volumes of data as they are reported by hospitals to the NRE. We will also include information in the studies regarding the increase in patient expectations, life expectancy and the prevalence of obesity. We will study the influence of the pandemic period on orthopedic surgeries and the return to post-pandemic normality [54,55].

## 5. Conclusions

This is the first research that, with the support of exhaustive data from the National Endoprosthetic Registry, analyzes the annual number of orthopedic surgeries in Romania for the period 2001−2022. During the studied period, 1,093,102 patients were operated on with various interventions: 189,881—hip replacement surgery; 51,035—knee replacement surgery; 11,085—revision hip arthroplasty; 1497—revision knee arthroplasty; 541,440—operated fractures; and 16,418—operated bone tumors.

The annual growth rate of hip replacement surgery is +8.19%, and the annual growth rate of knee replacement surgery is +19.55%. For primary arthroplasty interventions, there are clear gender differences. The annual growth rate for knee arthroplasty interventions is 2.38 times higher compared to hip arthroplasty. The proportion of women who underwent hip replacement surgery was 58.82%, and the proportion of women who underwent knee replacement surgery was 76.42%.

The annual growth rate of the number of revision hip arthroplasties is +9.43%, and the annual growth rate of the number of revision knee arthroplasties is +28.57%. The annual growth rate for knee revisions is 3.02 times higher for knee procedures compared to hip procedures.

Primary and revision hip volumes will double by 2034, while primary and revision knee volumes will double by 2027.

Modern treatments have led to an annual decrease of −4.52% in operated bone tumors.

The distribution of the number of primary and revision surgical interventions by county, compared to the population, shows consistent variations, the highest densities being recorded in the capital and the counties where university centers are located: Bucharest, Targu Mures, Timisoara, Cluj–Napoca and Brasov. Special regional performances are in Mures county, which has the highest densities of hip arthroplasties and the highest revision burden of the hip. This information is useful for identifying the best-performing hospitals in the country, collecting the best practices and making an international comparison of variation in performance between hospitals.

The results of this study allow further knowledge of the anticipated large increases in orthopedic surgery. This provides a quantitative basis for future policy decisions related to the number of orthopedic surgeons needed to perform these procedures, but also the allocation of adequate material resources to meet this need.

## Figures and Tables

**Figure 1 healthcare-11-01866-f001:**
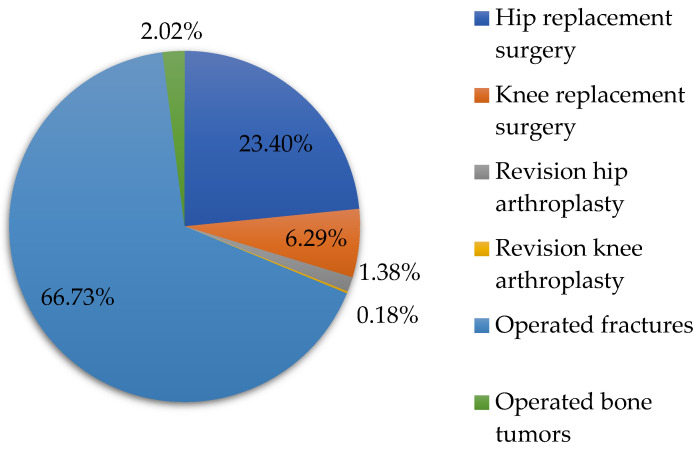
The percentage shares of orthopedic surgeries in the period 2001–2022.

**Figure 2 healthcare-11-01866-f002:**
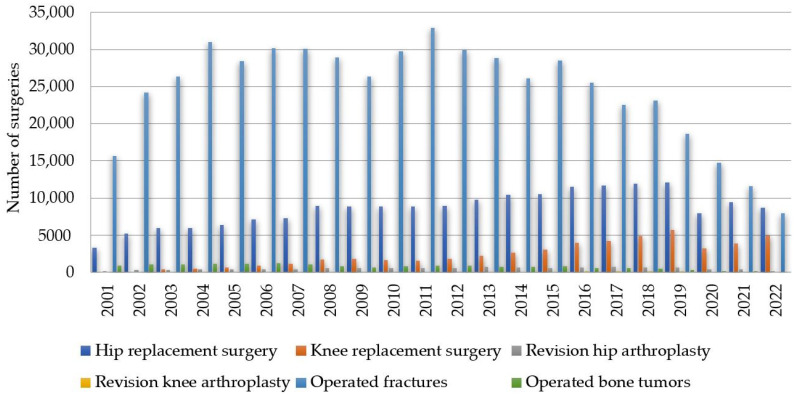
Prevalence of annual orthopedic surgeries between 2001 and 2022 in Romania based on NER for hip replacement surgery, knee replacement surgery, revision hip arthroplasty, revision knee arthroplasty, operated fractures and operated bone tumors.

**Figure 3 healthcare-11-01866-f003:**
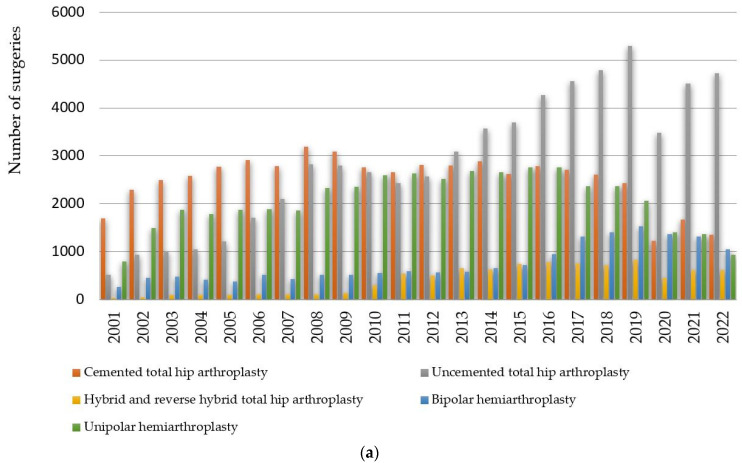
(**a**) Prevalence and (**b**) cumulative gender distribution of annual orthopedic surgeries between 2001 and 2022 in Romania based on RNE for hip replacement surgery: total hip arthroplasty (cemented total hip arthroplasty, uncemented total hip arthroplasty, hybrid and reverse hybrid total hip arthroplasty), bipolar hemiarthroplasty, unipolar hemiarthroplasty—Moore type.

**Figure 4 healthcare-11-01866-f004:**
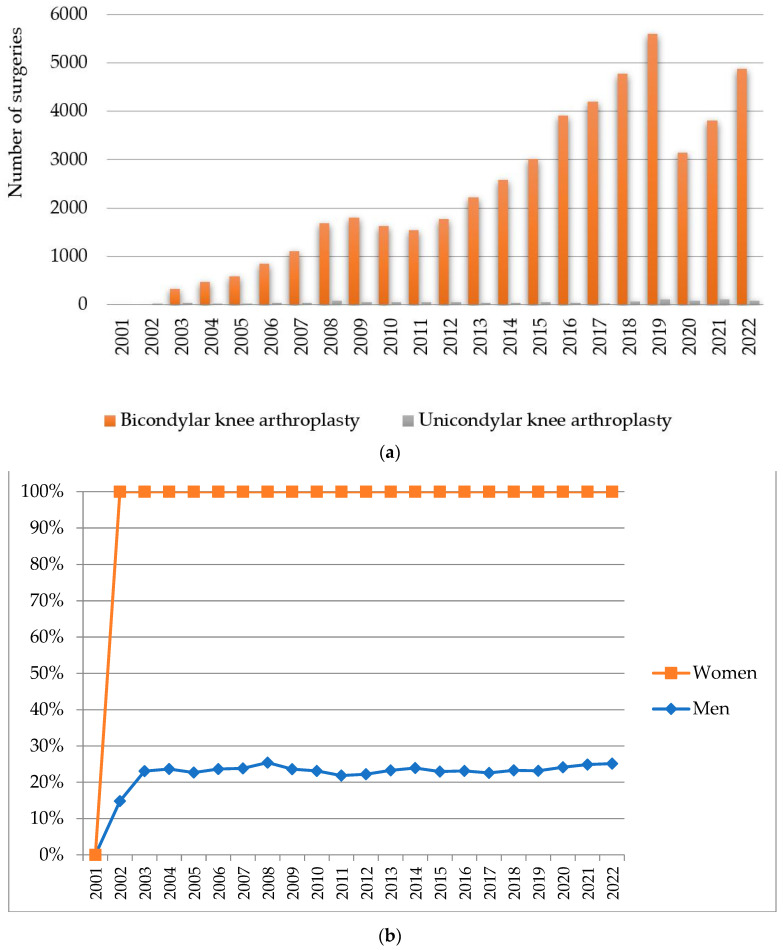
(**a**) Prevalence and (**b**) cumulative gender distribution of annual orthopedic surgeries between 2001 and 2022 in Romania based on NER for knee replacement surgery: bicondylar knee arthroplasty and unicondylar knee arthroplasty.

**Figure 5 healthcare-11-01866-f005:**
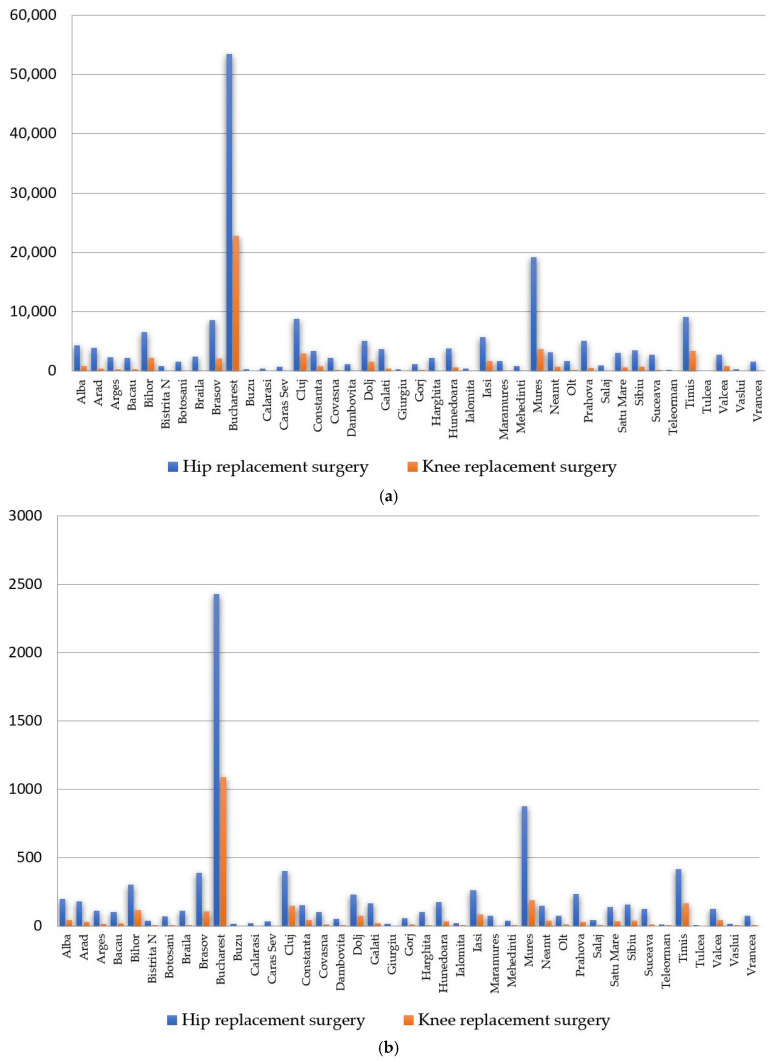
Hip and knee replacement surgeries in individual counties in period 2001–2022: (**a**) total numbers; (**b**) annual average.

**Figure 6 healthcare-11-01866-f006:**
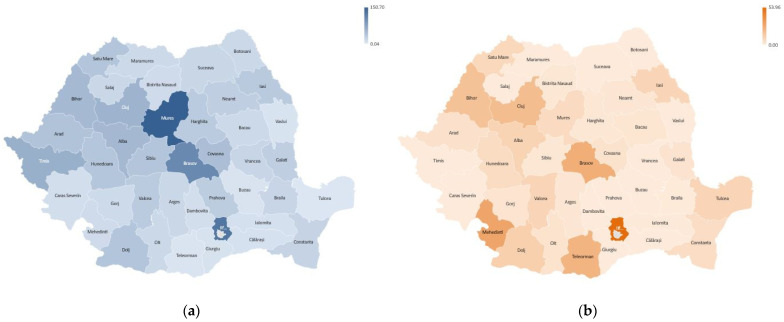
Density map by county in the period 2001–2022 for (**a**) hip replacement surgery; (**b**) knee replacement surgery.

**Figure 7 healthcare-11-01866-f007:**
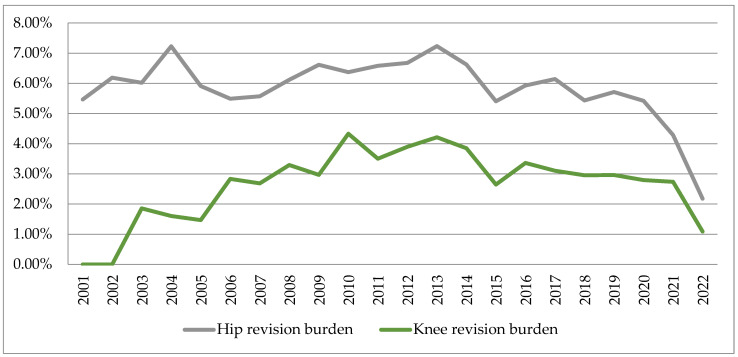
Variations of the hip and knee arthroplasties revision burdens at national level in the period 2001−2022.

**Figure 8 healthcare-11-01866-f008:**
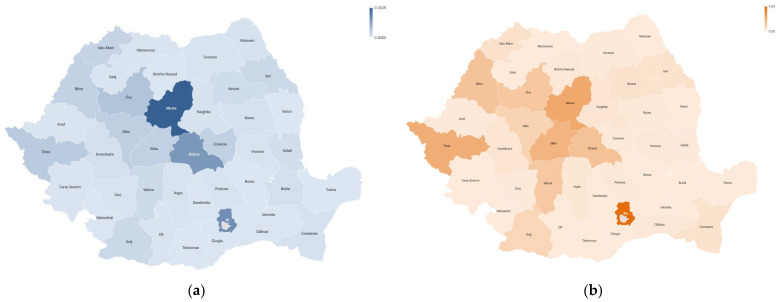
Revision burden maps by county in the 2001–2022 period for (**a**) revision hip arthroplasty; (**b**) revision knee arthroplasty.

**Table 1 healthcare-11-01866-t001:** Total number of patients in the period 2001–2022 who participated in the study.

Hip Replacement Surgery	Knee Replacement Surgery	Revision Hip Arthroplasty	Revision Knee Arthroplasty	Operated Fractures	Operated Bone Tumors	Hospitalized Patients	Operated Patients
189,881	51,035	11,085	1497	541,440	16,418	1,557,247	1,093,102

**Table 2 healthcare-11-01866-t002:** Variation trends of orthopedic surgical interventions in Romania.

Orthopedic Interventions	Variation Trends	
	Including Pandemic [%]	Without Pandemic [%]
Hip replacement surgery (Hrs)	5.89	8.19
Total hip arthroplasty (THA)	6.59	8.21
Cemented total hip arthroplasty (CTHA)	0.6	2.51
Uncemented total hip arthroplasty (UTHA)	13.05	15.22
Hybrid and reverse hybrid total hip arthroplasty (HTHA)	0.18	0.22
Bipolar hemiarthroplasty (BH)	0.09	0.12
Unipolar hemiarthroplasty—Moore type (UH)	0.03	0.07
Hip replacement surgery—male (HrsM)	0.07	0.09
Hip replacement surgery—female (HrsF)	0.05	0.08
Knee replacement surgery (Krs)	16.72	19.55
Bicondylar knee arthroplasty (BKA)	17.48	20.41
Unicondylar knee arthroplasty (UKA)	14.43	18.16
Knee replacement surgery—male (KrsM)	17.57	20.87
Knee replacement surgery—female (KrsF)	16.48	19.57
Revision hip arthroplasty (RHA)	3.46	9.43
Revision knee arthroplasty (RKA)	19.98	28.57
Operated fractures (Of)	−1.73	2.08
Operated bone tumors (Obt)	−8.95	−4.52
Hospitalized patients (Hp)	−4.33	−0.56
Operated patients (Op)	−1.96	−1.89

**Table 3 healthcare-11-01866-t003:** Arthroplasty revision burdens.

	2001	2002	2003	2004	2005	2006	2007	2008	2009	2010	2011
Hip											
Primary	3312	5236	5947	5922	6345	7141	7286	8974	8891	8869	8859
Revision	181	324	358	428	375	392	406	549	588	565	583
Revision burden	5.46	6.18	6.01	7.22	5.91	5.48	5.57	6.11	6.61	6.37	6.58
Knee											
Primary	0	27	377	499	613	883	1154	1761	1854	1685	1598
Revision	0	0	7	8	9	25	31	58	55	73	56
Revision burden	0	0	1.85	1.60	1.46	2.83	2.68	3.29	2.96	4.33	3.50
	**2012**	**2013**	**2014**	**2015**	**2016**	**2017**	**2018**	**2019**	**2020**	**2021**	**2022**
Hip											
Primary	8973	9814	10,401	10,546	11,554	11,705	11,893	12,134	7933	9471	8675
Revision	599	710	689	570	685	719	646	693	430	406	189
Revision burden	6.67	7.23	6.62	5.40	5.92	6.14	5.43	5.71	5.42	4.28	2.17
Knee											
Primary	1822	2255	2625	3061	3958	4222	4844	5704	3221	3908	4964
Revision	71	95	101	81	133	131	143	169	90	107	54
Revision burden	3.89	4.21	3.84	2.64	3.36	3.10	2.95	2.96	2.79	2.73	1.08

## Data Availability

The data used in this study can be requested from the corresponding author.

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
