# Peer review of "A Comprehensive Research on the Prevalence and Evolution Trend of Orthopedic Surgeries in Romania"

_healthcare, 2023, doi:10.3390/healthcare11131866_

Round 1
Reviewer 1 Report
I read with great interest your paper entitled "A Complete Research on the Prevalence and Evolution Trend of Orthopedic Surgeries in Romania"
The topic is interesting and reports the increase in TKA and THA rates during the study period. It also reports the geographical distribution.
Although it is something practical for Romanian health policymakers, I think it does not add significantly to the literature.
The TKA and THA rates are increasing as the elderly population is increasing. "In Romania, the proportion of population over 65 increased from 12.50% in 2000 to 14.94% in 2010. In 2020 the aged population will be 17.43 % and in 2030, 20.25%." As you yourself had hypothesized.
It is obvious that the geriatric population needs more arthroplasties. Larger centers and academic centers report more cases. And the geographical distribution is only notable for your own country members.
I think publication in a local journal is more suitable and you need to summarize the paper as it is too long for your data discussion and description.
Best,
minor lang. editing is needed.
Reviewer 2 Report
The paper entitled, `A Complete Research on the Prevalence and Evolution Trend of Orthopedic Surgeries in Romania'.
The following aspects need to be addressed:
Title
1. To change the title to A comprehensive Research on the Prevalence....
Abstract
2. The abstract is adequate and comprehensive.
Introduction
3. This section also is very clear and useful
4. Objectives of the studies are very clear.
Materials and methods
5. This section is very clear and easy to follow.
Results
6. It is recommended that the author can compare the result with other countries as well i.e. USA, or Australia.
Conclusion
7. The conclusion is very concise and precise.
References
8. There are 55 references used. The papers referred to are current and up-to-date.

Reviewer 3 Report
Dear Authors,
I read your paper with interest and appreciation. Your manuscript is interesting for people dealing with hip and knee arthroplasty and may be a cause for other countries to establish a national registry. In my opinion you should underline whether data collected in Romanian National Registry may also present the other problems like for instance: the type of implanted socket (hemispherical, threaded), the type of stem (surface replacement, short metaphyseal, standard), type of articulation (ceramic/ ceramic, metal/ polyethylene, etc.), the cause of revision etc. So, to summarize, does the registry give the possibility to observe trends not only in the most basic fields as you presented in the manuscript or also in more precise areas. Is it possible with the registry to present survival analysis of the implants?
Round 2
Reviewer 1 Report
Dear authors,
Your responses and revised version were convincing.
Best,